# Arthroscopic Medialization Partial Repair with Biologic Interposition Tuberoplasty for Large to Massive Irreparable Rotator Cuff Tear

**DOI:** 10.3390/medicina60030484

**Published:** 2024-03-14

**Authors:** Jae-Wook Jung, Joong-Bae Seo, Jun-Yeul Lee, Jae-Sung Yoo

**Affiliations:** Department of Orthopaedic Surgery, Dankook University College of Medicine, Cheonan 31116, Republic of Korea; jjw@dkuh.ac.kr (J.-W.J.); ssjb1990@dkuh.ac.kr (J.-B.S.); welivehospital@gmail.com (J.-Y.L.)

**Keywords:** rotator cuff, rotator cuff injuries, failure, acellular dermis

## Abstract

An irreparable rotator cuff tear is a challenging condition to treat, and various treatment modalities are being introduced. Medialization in the partial repair method has the limitation of exposing the tuberosity, while tension-free biologic interposition tuberoplasty using acellular dermal matrix has the limitation of exposing the humeral head. The authors believe that by combining these two techniques, it is possible to complement each other’s limitations. Therefore, they propose a surgical method that combines medialization and biologic interposition tuberoplasty for addressing these constraints.

## 1. Introduction

The origin of rotator cuff ailments is probably complex, involving a variety of factors, such as degeneration associated with age, as well as both minor and major injuries [1]. As individuals grow older, the likelihood of experiencing rotator cuff tears rises, with over half of people in their 80s exhibiting such tears [1]. Large to massive rotator cuff tears pose a challenging condition, and various treatment methods have been proposed [2,3,4]. If the retracted cuff in a large to massive cuff tear cannot be reduced to its anatomical position, partial repair, which repairs only the parts that can be repaired, can be a good option [5,6]. Partial repair is a convenient technique that can be performed without the use of additional synthetic tissue, yielding favorable outcomes [5,6]. However, it has the drawback of a high retear rate [5,6,7,8]. In an effort to overcome this, Lee et al. reported reducing the retear rate by medializing the footprint attachment site. They followed up with 42 patients with large to massive rotator cuff tears for 2 years and statistically significant clinical improvement was observed. However, we reported a retear rate of 23.8% [9]. Moreover, this procedure also has the limitation of not adequately covering and exposing the humerus greater tuberosity footprint [2,3,4,9]. The exposure of the footprint can contribute to ongoing impingement symptoms as impingement between the greater tuberosity and acromion remains unresolved.

Among various surgical procedures, superior capsular reconstruction is one of the most commonly employed techniques [10,11]. In order to address the drawbacks associated with the use of the Tensor Fascia Lata in the origin technique and to circumvent donor site issues, the method utilizing acellular dermis matrix has become increasingly popular in recent times. Satisfactory clinical and radiological outcomes have been reported with superior capsular reconstruction using acellular dermal matrix [2,11,12].

However, since superior capsular reconstruction connects the glenoid and humeral head, it cannot be completely tension-free, and the tension applied to the graft is closely related to the risk of retear [10,13]. Mirzayan et al. compared the retear patterns of superior capsular reconstruction and reported that when retears occurred at the humerus site, poor outcomes were observed. However, when retears occurred at the glenoid site, similar satisfactory clinical outcomes were achieved compared to the healed group [14,15]. In 2023, Mirzayan et al. have reported satisfactory outcomes by performing a biologic tuberosity interposition solely on the humerus tuberosity when graft failure occurs on the glenoid side. They based their findings on favorable results similar to patients with healed grafts [14,15]. However, a limitation of the biologic tuberosity is that it exposes the humerus head.

Therefore, as the application sites of medialization repair and biologic tuberosity are separate, combining these procedures is believed to complement the limitations of each technique. The authors aim to introduce this surgical method with the expectation that it can address the shortcomings of each procedure when used in combination.

## 2. Patients and Methods

Patients with massive irreparable tears but normal or repairable subscapularis were eligible for inclusion. Exclusion criteria comprised individuals with irreparable subscapularis, a previous history of infection, those diagnosed with joint arthritis categorized as Hamada stage 3 or higher, and post-operative infections [16]. Prior to the surgical intervention and during the follow-up post-surgery, the researchers evaluated the patients’ Visual Analog Scale pain scores, American Shoulder and Elbow Surgeons scores, range of motion, retear incidence, and acromio-humeral distance [17].

## 3. Surgical Technique

All patients underwent the application of regional anesthesia, and the procedures were performed in the beach chair position. After performing diagnostic arthroscopy (Arthrex, Naples, FL, USA), intraarticular single-row repair using the in-box technique was conducted up to Lafosse classification IV subscapularis tear. For Lafosse classification V subscapularis tear, a double-row repair was performed using the out-box technique [18].

To ensure comprehensive observation, debridement was carried out to eliminate any unhealthy tendon tissue and to enhance access to the tendon structures. The tear’s configuration was verified, and the potential presence of delamination was detected through arthroscopic examination. The extent of tendon mobility in both medial to lateral and posterior to anterior directions was carefully assessed using a grasping tool. If the mobility of a tendon was deemed inadequate for repair, procedures were undertaken to increase its mobility. Utilizing a shaver and radiofrequency ablation device, sufficient movement of the torn tendon was facilitated. Furthermore, intra-articular release of the tendon-capsular interface, which involved superior capsulotomy, was performed, along with releases of the tendon-bursal interface within the subacromial space.

When the biceps was preserved, anterior cable reconstruction using the long head biceps tendon was performed [4]. Biceps tenodesis is subsequently carried out at the superior aspect, 1 cm above the bicipital groove, using a percutaneous approach to insert a triple-loaded Y-Knot^®^ RC All-Suture Anchor (ConMed, New York, NY, USA). The initial stitch should penetrate the front portion of the long head biceps tendon to reposition the tendon towards the back. Next, another stitch encircles the long head biceps tendon before being secured. Lastly, a third stitch is threaded through the long head biceps tendon to ensure fixation at the anchoring location. By securing and stabilizing the long head biceps tendon at the anchoring point, it functions as an anterior cable reconstruction, extending from the anchor to the glenoid attachment site. From the anchoring point to the bicipital groove, the long head biceps tendon incorporates a biceps tendon interposition between the rotator cuff tear and the humeral head, promoting biological healing and reinforcing compromised rotator cuff tendons.

Subsequently, the medialization position was determined for the appropriate tension-free repair of the retracted cuff tendon. Medialization was performed when the length of the contact area between the tendon and bone, from medial to lateral, measured at least 1cm after the retracted tendon was pulled taut using a grasper in a tension-free manner. Single-row repair was conducted using 2–3 triple-loaded Y-Knot^®^ RC Anchors (ConMed, New York, NY, USA). Following this, the most anterior and posteriorly tied FiberWire strands were left in place for biologic tuberoplasty (Figure 1).

Subsequently, the anterior-posterior and medial-to-lateral distances of the exposed tuberosity were measured. After medialization, the remaining gap was addressed by securing the medial side of a 4mm-thick acellular dermal matrix (BellaCell; Hans Biomed Corporation, Daejeon, Republic of Korea). After securing the medial side of the acellular dermal matrix, the remaining sutures were effectively anchored to the greater tuberosity using two PopLock or Argo knotless anchors (ConMed, New York, NY, USA) via a suture bridge technique [17] (Figure 2). 

## 4. Postoperative Management

All patients wore a shoulder abduction brace for 6 weeks post surgery. Pendulum exercises commenced at 1week post-surgery, and passive joint exercises were gradually initiated from 2 weeks post-surgery, with restrictions ensuring no more than 120 degrees of forward elevation and 30 degrees of external rotation until 6 weeks post-surgery. Assisted active joint exercises began at 6 weeks post-surgery, followed by active joint exercises at 8 weeks post-surgery, and resistive strengthening exercises at 12 weeks post-surgery. Progressive open and closed chain exercises were implemented, allowing for gradual return to sports activities by the 6-month postoperative mark.

## 5. Result

### Case 

A 71-year-old female patient presented with persistent pain and muscle weakness over several months, leading to a surgical consultation. Preoperative magnetic resonance imaging revealed a massive rotator cuff tear (Figure 3A). Surgical intervention involved medialization rotator cuff partial repair and biologic interposition tuberoplasty using acellular dermal matrix. Postoperative magnetic resonance imaging demonstrated full coverage from the humeral head to the tuberosity (Figure 3B).

At the 6-month follow-up, the patient exhibited Visual Analog Scale pain score is 1, American Shoulder and Elbow Surgeons score is 90 and regained full range of motion (Figure 4).

## 6. Discussion

The rotator cable protects the rotator crescent, which is where the majority of rotator cuff tears originate, by shielding it from stress [19,20]. Similar to the cables found in suspension bridges, it enables the supraspinatus to transfer its muscle force to the humerus, even when a tear is present [21]. Hence, rotator cuff tears affecting the anterior or posterior attachment of the rotator cable are believed to induce heightened stress on the crescent tissue, thereby facilitating the progression of the rotator cuff tear [22]. In 2014, Namdari et al. observed that the rupture of the anterior supraspinatus tendon correlated with larger tear sizes and more severe supraspinatus muscle degeneration in cases of painful small- and medium-sized rotator cuff tears [23]. Opting for the long head biceps tendon as an autograft source for anterior cable reconstruction is rational and entails minimal patient morbidity, considering its close proximity to the rotator cuff. Moreover, concurrent long head biceps tendon pathology frequently occurs in the context of rotator cuff tears, and tendon sacrifice through tenotomy or tenodesis is a standard practice [19,20,24,25]. Seo et al. reported that anterior cable reconstruction with the long head of the biceps tendon demonstrated favorable clinical and radiologic outcomes compared to the conventional rotator cuff repair-only technique. This approach prevented retears following rotator cuff repair and enhanced the acromiohumeral distance. However, there was no discernible difference in clinical outcomes between the two groups [4].

Large to massive rotator cuff tears are known to be challenging to treat due to chronic tendon wear, severe retraction, and significant fatty degeneration of the muscles [26,27]. It is known that attempting to repair severely retracted tendons with excessive tension is closely associated with a high rate of retear [5]. Yoo et al. described that inadequate coverage of the original greater tuberosity footprint during arthroscopic repair of large to massive rotator cuff tears was linked to a relatively elevated retear rate, standing at 45.5% [5]. In a review article published in 2022, an analysis of eighty-two studies encompassing a follow-up of 2790 shoulders was conducted. Across all procedures, there was an improvement in shoulder scores in the short term. However, at the 2-year mark post-surgery, the groups undergoing balloon spacers, arthroscopic debridement, and partial cuff repair exhibited unfavorable outcomes in shoulder scores. Additionally, high rates of retear were reported in partial cuff repairs (45%), and superior capsule reconstruction (21%) [28]. Therefore, methods aimed at reducing tension, such as footprint medialization repair, have been reported to decrease retear rates [9,29]. However, although medialization partial repair reduces retear rates, complete prevention of retear is still not possible. Lee et al. reported a retear rate of 23.8% after conducting medialization and following up with 42 patients over a period of 2 years [9]. Additionally, because exposure of the humerus greater tuberosity cannot be addressed, preventing bone-to-bone contact between the greater tuberosity and acromion, there is a drawback in that pain relief or functional improvement cannot be expected.

Allodermal graft utilization in superior capsular reconstruction is favored and extensively embraced because it circumvents the donor site complications linked with autograft superior capsular reconstruction utilizing the tensor fascia lata, thereby minimizing donor site morbidity [2,12]. Nonetheless, superior capsular reconstruction presents a limitation in which the surgical attachment of the allodermis graft from the scapular glenoid to the humeral greater tuberosity induces tension on the graft, posing challenges in preventing graft failure [14,15]. Mirzayan et al. detailed that instance of graft failure yet preserved humeral coverage yielded comparable satisfactory outcomes to patients with intact grafts. Conversely, patients experiencing tears on the tuberosity side, leading to loss of tuberosity coverage, exhibited unfavorable results [14,15].

Mirzayan et al. introduced biologic interposition tuberoplasty to prevent inevitable retears occurring in superior capsular reconstruction and reported satisfactory outcomes [14,15]. In 2023, they conducted biologic interposition tuberoplasty on 12 patients, reporting improved pain and shoulder scores postoperatively. Among the 12 patients, MRI was performed at an average of 5.3 months after surgery in 7 cases, revealing healing without any instances of retear in all cases [14,15]. In 2023, Seo et al. also reported a retear rate of 0% by simultaneously performing biologic interposition tuberoplasty and bursal acromial reconstruction. They observed prevention of bone-to-bone contact between the greater tuberosity and acromion, along with an increase in acromiohumeral distance, resulting in pain relief and functional improvement [17]. However, one drawback of biologic tuberoplasty is that it positions tissue from the medial aspect of the footprint rather than from the glenoid, leaving the medial aspect of the humeral head exposed. Additionally, for severely retracted tendons, adhesiolysis and medialization often allow for adequate repair. Since the repair locations of medialization and biologic tuberosity do not overlap and complement each other in terms of exposed areas, both techniques were applied together, resulting in complete resolution of humeral exposure post-surgery. 

Partial repair with medialization is a straightforward procedure, but it carries the risk of retear and ongoing impingement due to humeral head exposure. superior capsular reconstruction allows for tissue interposition from the glenoid to the humeral head without exposing them fully, but it is susceptible to retear due to tension. Biologic tuberoplasty is a tension-free surgical procedure, minimizing the risk of retear; however, it does not address the exposure on the medial side of the humeral head. We have proposed a method of combining partial repair with medialization and biologic tuberoplasty. The first advantage is that the anatomical locations of the procedures do not overlap, making the surgery easier. The second advantage is that by achieving tendon healing through partial repair with medialization, impingement symptoms that may occur at the medial side of the humeral head can be improved, and functional recovery can be expected due to tendon healing. Furthermore, even if the partially repaired tendon experiences a retear, the tension-free interposition tuberosity fixed independently on the lateral side maintains functionality without causing impingement. Thus, we can anticipate improvements in pain and function. Therefore, it is anticipated that the applicability of this approach will be quite broad, depending on the extent of retracted cuff tension.

The authors presented a table outlining the surgical targets, limitations, and retear risks of the four procedures: partial repair with medialization, superior capsular reconstruction, biologic tuberoplasty, and the combination of medialization and biologic tuberoplasty. They suggested that clinical and radiological studies comparing these procedures should be provided in the future (Table 1) [3,9,13,17].

## 7. Conclusions

In summary, the combined use of medialization and biologic interposition tuberoplasty serves as a surgical approach that complements the limitations of each technique. Due to the anatomical improvements in various areas compared to standalone procedures, better pain relief and functional enhancement are anticipated.

## Figures and Tables

**Figure 1 medicina-60-00484-f001:**
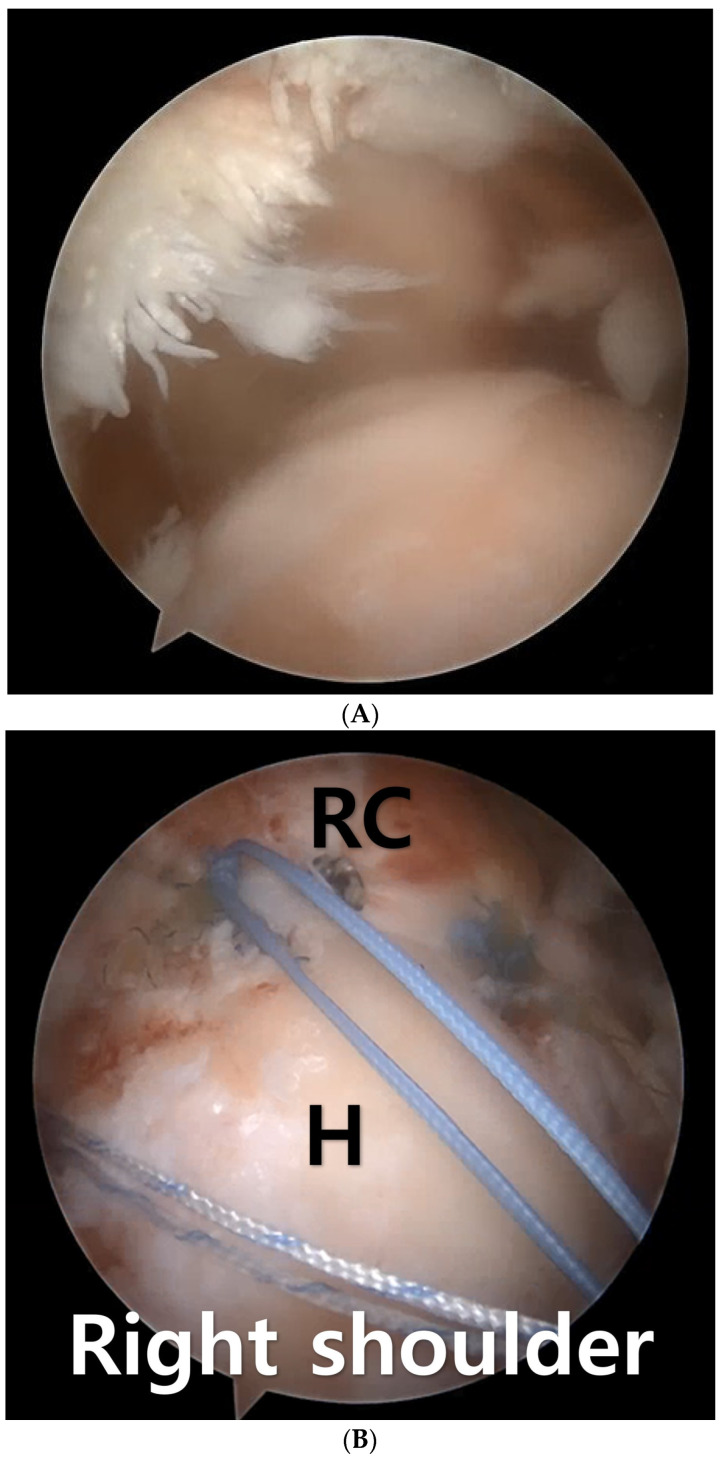
(**A**) The arthroscopic examination of a 72-year-old female reveals findings consistent with a massive rotator cuff tear. (**B**) Footprint medialization and single-row repair were performed using two suture anchors, and three pairs of FiberWire sutures were left for the biologic tuberoplasty procedure. RC; rotator cuff, H; Humerus.

**Figure 2 medicina-60-00484-f002:**
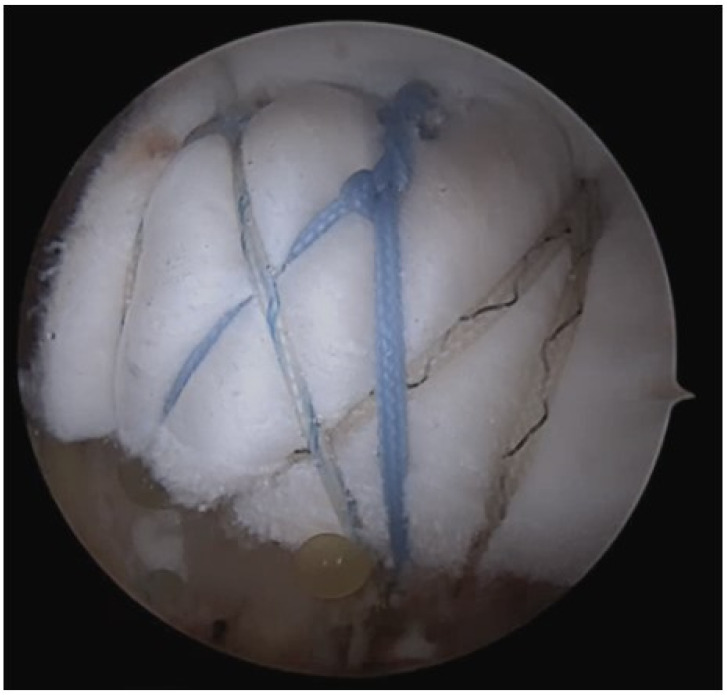
The remaining FiberWire sutures from the medial row and acellular dermal matrix were interposed, and biologic tuberoplasty was performed using the suture bridge repair method.

**Figure 3 medicina-60-00484-f003:**
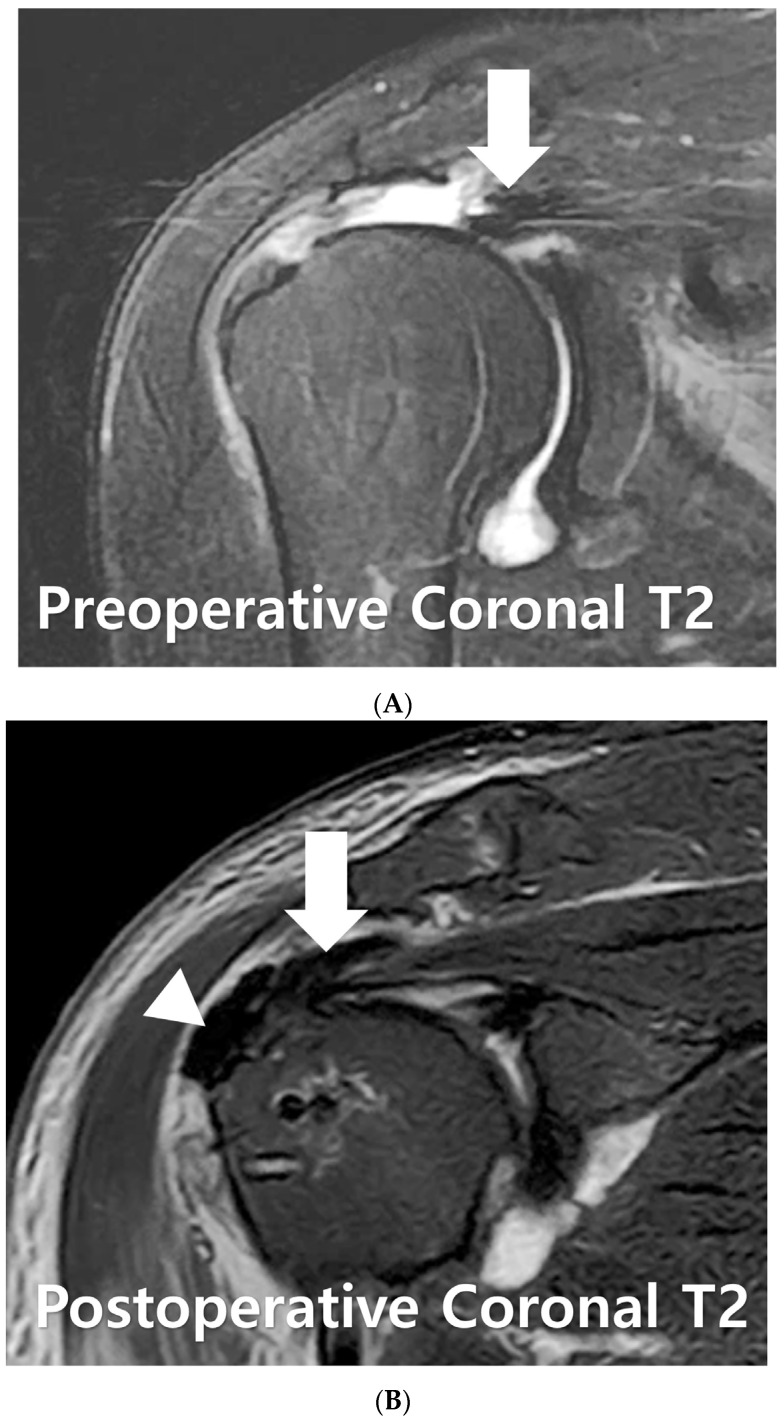
(**A**) On preoperative magnetic resonance imaging of a 72-year-old female with a massive rotator cuff tear, findings reveal a large-sized acromial spur and retracted cuff tear (arrow). (**B**) Postoperative magnetic resonance imaging shows the medialization and repair of the cuff tendon (arrow), as well as the presence of biologic interposition tuberoplasty (arrowhead).

**Figure 4 medicina-60-00484-f004:**
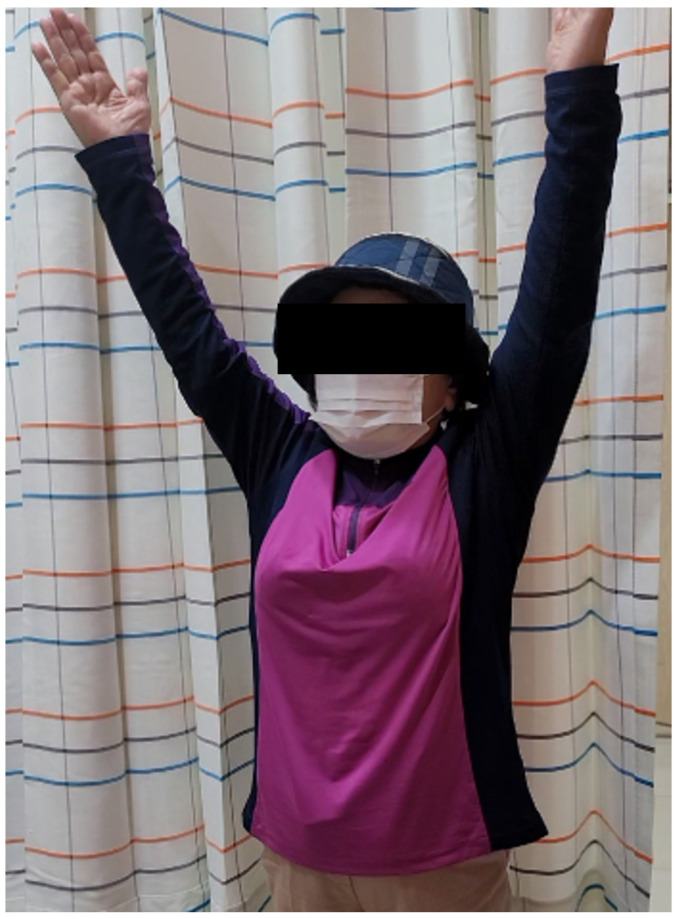
A satisfactory outcome was observed at 6 months post-surgery, with the patient demonstrating active full range of motion.

**Table 1 medicina-60-00484-t001:** Comparative analysis of four surgical procedures.

	Medialization	SCR	BT	Medialization + BT
Repair area	GT medial side	Glenoid~GT	GT	All
Exposure area	GT	None	GT medial side	None
Retear rate	High	Moderate	Low	Low

SCR; superior capsular reconstruction, BT; Biologic tubeoplasty, GT: greater tuberosity.

## Data Availability

It is available when reviewers request.

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
