# Peer review of "Arthroscopic Medialization Partial Repair with Biologic Interposition Tuberoplasty for Large to Massive Irreparable Rotator Cuff Tear"

_medicina, 2024, doi:10.3390/medicina60030484_

Round 1
Reviewer 1 Report
Comments and Suggestions for Authors
Please check the manuscript for the abbreviations. Please correct where necessary.
Every statement has to be supported by specific references.
Please provide the number and date of the ethics committee approval.
The manuscript should be improved as Technical note should include an Abstract, Keywords, Introduction, Materials and Methods, Results, Discussion, and Conclusions.
Keywords
They have to be better focused on the topic.
Introduction
Introduction has to be improved better describing the surgical procedures and their drawbacks.
Please update the references to the most recent ones. Reporting systematic reviews could be more useful.
Lines 16- 18. Authors reported “Among them, partial repair is a convenient technique that can be performed without the use of additional synthetic tissue, yielding favorable outcomes.“ Please add specific references.
Line 18: please substitute reference 4 with a more recent and specific one also reporting comparison of outcomes of different techniques.
Line 21: humerus great tuberosity is first mentioned here. Please use the abbreviation GT and check the whole manuscript.
Lines 22-23: please add a specific reference.
Lines 22-25: please better describe the reported techniques along with their specific drawbacks. Please add specific references.
Line 28: please add a specific reference.
Patients and Methods
This section is lacking. Please add.
Please add the preoperative evaluations and indications.
Please specify the inclusion and exclusion criteria.
Clinical scores are lacking. Please add.
Surgical Technique
Line 39: Please add the complete information (name, city, country, etc.) about the arthroscope used.
Lines 40-42. Please describe the Lafosse classification also adding a specific reference.
Line 46: Please add the complete information (name, city, country, etc.) about the FiberWire strands.
Case
Lines 75-76: acellular dermal matrix has already been abbreviated. Please correct.
Results are lacking.
Discussion
The discussion should be improved
The reported studies should be better described and discussed.
Line 92: please add specific references.
Line 100: Superior Capsular Reconstruction has already been abbreviated. Please correct.
Line 103: greater tuberosity has already been abbreviated. Please correct.
Line 114: please clarify the postoperative risks.
Conclusion
This section is lacking. Please add.
Figures
Figure 1, 3 and 4: (A) and (B) are not reported in the figure. Please add.
Figure 2: Please add rows to facilitate the reader.
Table 1
Table has to be improved reporting data and references about the different surgical procedures.
Authors reported that “This suggests that compared to SCR or medialization alone, biologic tuberosity can mitigate the risk of re-tear and potentially lead to better functional improvement by promoting healing of the medialized tendon”. This data has to be reported.
References
References from nr 13 to nr 17 are not reported in the manuscript. Please add.
Comments on the Quality of English LanguageEditing of English language and style is required.
Author Response
Answer to reviewer’s comments
I deeply appreciate the valuable insights and efforts of the reviewers and editors on our paper. They have greatly contributed to the growth and advancement of our research
Introduction has to be improved better describing the surgical procedures and their drawbacks.
Please update the references to the most recent ones. Reporting systematic reviews could be more useful.
àI have made the requested additions regarding the surgical technique and drawbacks in the introduction section. Additionally, I have included recent systematic review articles in the references as requested. Once again, thank you for your guidance.
Lines 16- 18. Authors reported “Among them, partial repair is a convenient technique that can be performed without the use of additional synthetic tissue, yielding favorable outcomes.“ Please add specific references.
àThank you for the valuable feedback. I have added the references as suggested to the relevant sentences.
Line 18: please substitute reference 4 with a more recent and specific one also reporting comparison of outcomes of different techniques.
àThank you for the insightful suggestion. As requested, I have added recent papers comparing the technique with others as references. Your guidance is appreciated.
Line 21: humerus great tuberosity is first mentioned here. Please use the abbreviation GT and check the whole manuscript.
à Thank you for the precise feedback. I have clarified the abbreviation 'GT' as suggested.
Lines 22-23: please add a specific reference.
à Thank you for the helpful suggestion. I have elaborated on the surgical technique and drawbacks as requested, and I have also added references accordingly.Line 28: please add a specific reference.
Lines 22-25: please better describe the reported techniques along with their specific drawbacks. Please add specific references.
àThank you for the helpful suggestion. I have elaborated on the surgical technique and drawbacks as requested, and I have also added references accordingly.Line 28: please add a specific reference.

Reviewer 2 Report
Comments and Suggestions for Authors
The technical note is intriguing. are my suggestions:
Firstly, it would be beneficial to include the definition of partial repair in the introduction.
Secondly, consider providing information on the prevalence of rotator cuff tendon tears.
Thirdly, in the figures, I recommend adding labels to clarify the anatomical landmarks.
Lastly, for the MRI figures, please include image sequences and specify the imaging planes in the legends for better understanding.
Author Response
The technical note is intriguing. are my suggestions:
I deeply appreciate the valuable insights and efforts of the reviewers and editors on our paper. They have greatly contributed to the growth and advancement of our research
Firstly, it would be beneficial to include the definition of partial repair in the introduction.
à I appreciate the precise feedback. As suggested, I have elaborated on the definition of partial repair in the introduction section. Thank you for your input
Secondly, consider providing information on the prevalence of rotator cuff tendon tears.
à I appreciate the insightful feedback. As requested, I have provided additional details regarding the prevalence of rotator cuff tears. Thank you for your guidance
Thirdly, in the figures, I recommend adding labels to clarify the anatomical landmarks.
à I am grateful for the valuable suggestion. As pointed out, I have applied labels to anatomical structures within the figure. Thank you for your guidance
Lastly, for the MRI figures, please include image sequences and specify the imaging planes in the legends for better understanding.
à As suggested, I have provided additional explanations to aid in understanding the MRI images. I apologize for the inconvenience, but adding additional images is currently difficult due to recent job changes. Thank you for your understanding

Round 2
Reviewer 1 Report
Comments and Suggestions for Authors
Authors have to reply to all the suggestions. Some issues are still pending.
Authors should add line numbers to facilitate the revision of the manuscript.
Authors reported the following abbreviations:
Abbreviations: ABT: Arthroscopic biologic tuberoplasty, ACR; anterior cable recon-struction, AHD; acromiohumeral distance, ADM; acellular dermal matrix, MRCT: mas-sive rotator cuff tear, SCR: superior capsule reconstruction, MRI; magnetic resonance im-aging, BT; Biologic tuberoplasty, GT; greater tuberosity.
Some abbreviations are not always used. Please check the whole manuscript and correct.
Authors reported “re-tear” or “retear”. Please use only one definition. Please check the whole manuscript.
Keywords
They have to be better focused on the topic also based on the title of the manuscript.
Page 2
Authors reported Hamada classification for joint arthritis. Please clarify also adding specific references.
Authors reported “the researchers evaluated the patients' Visual Analog Scale pain scores, American Shoulder and Elbow Surgeons scores, range of motion, retear incidence, and acromio-humeral distance. Please clarify and add specific references.
Table 1
Table has to be improved reporting data and references about the different surgical procedures.
Authors reported that “This suggests that compared to SCR or medialization alone, biologic tuberosity can mitigate the risk of re-tear and potentially lead to better functional improvement by promoting healing of the medialized tendon”. This data has to be reported.
Figures
Figure 1 and 3: (A) and (B) are not reported in the figure. Please add.
References
Reference 23 is incomplete. Please correct
Comments on the Quality of English LanguageMinor editing of English language required
Author Response
Thank you to the judges and editorial board members for your hard work
Authors reported the following abbreviations:
Abbreviations: ABT: Arthroscopic biologic tuberoplasty, ACR; anterior cable recon-struction, AHD; acromiohumeral distance, ADM; acellular dermal matrix, MRCT: mas-sive rotator cuff tear, SCR: superior capsule reconstruction, MRI; magnetic resonance im-aging, BT; Biologic tuberoplasty, GT; greater tuberosity.
Some abbreviations are not always used. Please check the whole manuscript and correct.
à Thank you for the precise feedback. I have made the necessary revisions as indicated. (line 245-247)
Authors reported “re-tear” or “retear”. Please use only one definition. Please check the whole manuscript.
à Thank you for the precise feedback. I have made the necessary revisions as indicated. (line 29-32)
Keywords
They have to be better focused on the topic also based on the title of the manuscript.
à Thank you for the precise feedback. I have made the necessary revisions as indicated. (line 17)
Page 2
Authors reported Hamada classification for joint arthritis. Please clarify also adding specific references.
àThank you for the insightful comment. I have added a reference to the point you mentioned.(line 62)
Authors reported “the researchers evaluated the patients' Visual Analog Scale pain scores, American Shoulder and Elbow Surgeons scores, range of motion, retear incidence, and acromio-humeral distance. Please clarify and add specific references.
àThank you for the insightful comment. I have added a reference to the point you mentioned.(line 66)
Table 1
Table has to be improved reporting data and references about the different surgical procedures.
Authors reported that “This suggests that compared to SCR or medialization alone, biologic tuberosity can mitigate the risk of re-tear and potentially lead to better functional improvement by promoting healing of the medialized tendon”. This data has to be reported.
àThank you for the insightful comment. I have added a reference to the point you mentioned. (line 236)
Figures
Figure 1 and 3: (A) and (B) are not reported in the figure. Please add.
à
As you suggested, I have made the necessary revisions and annotated the diagram accordingly.
References
Reference 23 is incomplete. Please correct
à Thank you for the precise feedback. I have made the necessary revisions as indicated (line 308)